# Reinforcement Learning with Synthetic Navigation Data Allows Safe Navigation in Blind Digital Twins

## Abstract

Limited access to dedicated navigation data in visually impaired individuals is a significant bottleneck in the development of AI-driven assistive devices. To address this, we have developed a virtual environment designed to extract various human-like navigation data from procedurally generated labyrinths. Using reinforcement learning and semantic segmentation, we trained a convolutional neural network to perform obstacle avoidance from synthetic data. Our model outperformed state-of-the-art backbones including DINOv2-B in safe pathway identification in real world. In conclusion, despite being trained only on synthetic data, our model successfully extracted features compatible with safe navigation in real-world settings, opening new avenues for visually impaired.

## 1    Introduction

Globally, millions of people live with blindness, which is associated to severe restrictions in mobility and as well as a significant increased risk of fall-related injuries (Wood et al., 2011; Brunes & Heir, 2021; Singh & Maurya, 2022). The recent development of retinal and cortical prostheses has not achieved vision restoration in blinds yet. As an alternative, Sensory Substitution Devices (SSDs) hold significant promise for aiding visually impaired individuals by converting environmental sensor data into tactile or auditory stimuli (Bach-y Rita, 1972; Jicol et al., 2020). However, current SSDs face limitations in conveying complex visual scenes through skin or ears (Elli et al., 2014), primarily due to the narrow bandwidth of these sensory channels, which can lead to cognitive burden (De Jong, 2010). Indeed, artificial intelligence (AI), and in particular deep learning, enables the extraction of relevant information to be conveyed to the blind.

Contrary to autonomous driving (Chen & Krähenbühl, 2022; Toromanoff et al., 2019) and robotics (Shah & Levine, 2022; Kruse et al., 2013), SSDs have not benefited from the availability of navigation datasets. As a result, current applications of deep learning for effective navigation aids predominantly rely on general-purpose datasets for object recognition and classification (Scalvini et al., 2023; Mukhiddinov & Cho, 2021; Kim et al., 2023; Kerdegari et al., 2016), semantic segmentation (Tapu et al., 2017; Zheng & Weng, 2016), or depth estimation (Bai et al., 2017; Sharma et al., 2016; Asiedu Asante & Imamura, 2023). While these applications can extract high-level features that improve environmental understanding for the visually impaired, they do not convey information about navigation decisions. A few studies have used navigation-specific data to provide guidance (Zheng & Weng, 2016; Kerdegari et al., 2016), but the amount of data was limited to a few hundred samples due to the resource-intensive nature of the collection process.

The aim of this study is twofold: (i) To address the unavailability of human navigation data, we propose a generic method for training AI systems specifically designed for the blind. Our approach relies on synthetic semantic segmentation maps to optimize SSD outputs in virtual environments, enabling straightforward real-world transferability without requiring advanced domain adaptation (Xu et al., 2022; Zhu et al., 2023). (ii)

We evaluate the efficacy of this method by first training an AI-based system for obstacle avoidance in virtual environments and then demonstrating its ability to enable safe navigation using low-dimensional navigation cues in both virtual and real-world settings. To support this, we introduce NavIndoor, a new virtual environment designed for the automatic generation of synthetic human-like navigation data. NavIndoor leverages procedural generation to create randomized, obstacle-filled mazes with structure similar to real-world indoor environments, allowing for the simulation of various navigation scenarios in an efficient, safe, and scalable manner.

In Section 3, we describe our method for optimisation of AI-based SSDs within virtual environments and deployment in real-world setting. In Section 4, we present the newly proposed virtual environment and demonstrate that a compact CNN allows safe navigation in it. Finally, in Section 5, we illustrate the model's proficiency in real-world settings by estimating the forward navigation boundary in the Active Vision Dataset and showing that our model outperforms standard backbones in safe pathway identification (AUC 0.92). Furthermore, we show that incorporating random morphological operators around obstacles during training in virtual environments improves generalization to real-world data. The scalable and flexible nature of our method, combined with its potential for generalization to various navigation tasks, underscores its promise in enhancing feature extraction for future sensory substitution devices.

We summarize our contributions as follows:

- We identify a significant gap in the literature regarding the use of navigation data to improve Sensory Substitution Systems.
- We release NavIndoor, an open-source software for the computationally efficient generation of procedurally generated, obstacle-filled environments, enabling seamless integration with AI systems. NavIndoor facilitates the efficient creation of various large-scale, human-like navigation datasets.
- We show that synthetic data enables the extraction of low-dimensional features for navigation by individuals with visual impairments.
- We demonstrate that applying basic morphological operators to synthetic semantic segmentation maps enhances performance in real-world conditions after training.

## 2 RELATED WORKS

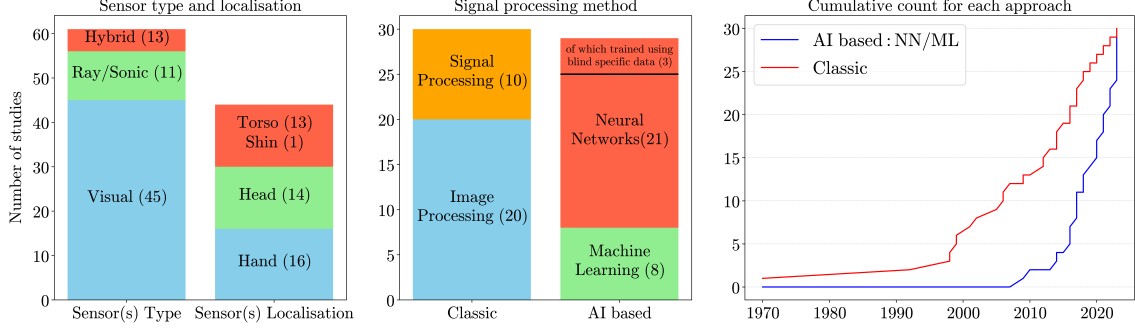

Figure 1: Overview of Sensory Substitution Devices Publications. (Left) Sensor type and their body localisation. (Middle) Processing methods used in the studies for classic and AI-based approaches. (Right) Cumulative count of publications over the years for machine learning (ML) or neural networks (NN) and classic approaches.

SENSORY SUBSTITUTION DEVICES AND AI FOR THE VISUALLY IMPAIRED

We reviewed the literature on sensory substitution devices and electronic travel aids released from 1970. The literature review was performed in PubMed and completed with research using arXiv, Elicit, and conventional search engines. Papers proposing new systems for sensory substitution were included, resulting in 61 studies. Detailed methodology and results are presented in Appendix A.

Various SSDs have been proposed to improve navigation for the blind. Traditional visual SSDs often utilized image mapping through haptic feedback (Bach-y Rita et al., 1998; Kajimoto et al., 2004; 2006; DANILOV & TYLER, 2005) or audio representations derived from images (Auvray et al., 2007; Meijer, 1992a). However, these approaches demand substantial attentional resources (Lee, 2019; Theurel et al., 2013), leading to cognitive overload (De Jong, 2010), and limiting their application to controlled environments (Elli et al., 2014).

Consequently, image and signal processing have been used to improve extraction of salient features using infrared, LiDAR, ultrasonic and mostly (50/61, 82%) visual sensors.

The use of machine learning and neural networks were introduced in 2009 and changed drastically the design of new devices by involving 58% (28/48) studies since then. Applications of neural networks for sensory substitution devices include image segmentation, object recognition, or classification (Busaeed et al., 2022; Scalvini et al., 2023; Asiedu Asante & Imamura, 2023; Afif et al., 2020; Sulaman et al., 2023; Bhatlawande et al., 2022; Chaudhary & Dr. PrabhatVerma, 2023; Mukhiddinov & Cho, 2021), object tracking (Tapu et al., 2017), speech understanding (Bai et al., 2017), image captioning (Ganesan et al., 2022; Kavitha et al., 2023), optimization of auditory representation of images with GANs (Kim et al., 2021; 2023; Hu et al., 2019; Port et al., 2021), and best action prediction (Zheng & Weng, 2016; Kerdegari et al., 2016).

AI-BASED DEVICES FOR OBSTACLE AVOIDANCE

Out of 20 neural-network SSD studies, 9 were aimed at obstacle avoidance tasks, which is critical for safe navigation. Limitations with such approaches mainly rely on computational and energetic cost, because such systems often require substantial hardware resources to perform multiple scene understanding tasks in parallel and in real-time (Mahendran et al., 2021). Also, complex operations such as depth estimation may require expensive or heavier sensors, such as stereo camera (Caraiman et al., 2017; Asiedu Asante & Imamura, 2023).

On the other hand, 2 studies (Zheng & Weng, 2016; Kerdegari et al., 2016) proposed to estimate directly the best possible action for the blind, but such tasks require extensive and costly acquisition of human navigation data. In (Zheng & Weng, 2016), authors collected 4109 tuples of GPS/visual sensor information labelled with the best possible action (forward/left/right/stop) predicted with a deep neural network. In (Kerdegari et al., 2016), authors collected 4051 samples comprising an ultrasonic measurement coupled with a performed action (forward/left/right) predicted with a multilayer perceptron.

DOMAIN ADAPTATION FROM SIMULATION TO REAL ENVIRONMENTS

Domain shift is a primary concern when deploying deep learning models trained in simulations into the real-world. Approaches to enable the transferability of AI systems from virtual to real-world environments have primarily focused on achieving photorealism by scanning real 3D scenes, as highlighted in (Xia et al., 2018). This strategy allows AI systems to utilize material textures for executing complex tasks and to leverage detailed 3D scene representations stored in external memory. Such systems have primarily benefited robotics (Hirose et al., 2019; Kang et al., 2019), where they enable fully autonomous agents to undertake complex scene understanding tasks.

Besides, SSDs are designed to prioritize the transmission of low-dimensional features that can be computed in real-time and easily interpreted by humans for navigation, and thus do not necessarily need to store a 3D rich representation of the surrounding environments for complex autonomous navigation tasks. Indeed, on the other hand, semantic segmentation has been proposed as domain-agnostic features to allow better generalization for robotics navigation in real-world (Hong et al., 2018; Chaplot et al., 2020). In a similar manner, we propose to use a simple semantic view of the scene as an input for bridging domain gap. However our approach is not designed to be deployed on fully autonomous agents, and focuses on extraction of low-dimensional navigation cues that could be interpreted by humans with haptic or auditory feedback. This approach enables the model to learn better obstacle avoidance as shown in Section 4, and considerably reduces the domain gap with real-world data. Compared with photorealism approaches, the use of semantic views also reduces the input dimensionality providing faster training and avoiding texture biases.

Figure 2: Method overview : we leverage training in virtual environments with reinforcement learning, by equipping a digital twin with sensor(s) to master navigation tasks from domain-invariant features. Post-training, utilizing the acquired knowledge encoded in $Q_\theta$ model, we extract navigation features from real-world data. Such features can finally be conveyed through a sensory substitution device (SSD). Virtual environment elements are denoted in blue, real-world elements in green, while cross-domain elements are highlighted in yellow.

EMBODIED AI PLATFORMS FOR ROBOTICS

AI-embodied platforms have primarily been developed for robotics navigation tasks, utilizing either synthetic assets or the scanning of real-world scenes. Scanning real-world scenes has led to the development of tools including Gibson (Xia et al., 2018; 2019), Habitat (Szot et al., 2021; Ramakrishnan et al., 2021), and Openroom (Li et al., 2021), providing hundreds of virtual scenes for training. Unity-based environments have also been developed by the Allen Institute for AI (AI2) (Ehsani et al., 2021; Deitke et al., 2020; 2022).

As proposed in ProcTHOR (Deitke et al., 2022), NavIndoor leverages procedural generation as a key element. In the context of mobility assistance, SSDs are expected to function in a wider range of environments

than those typically encountered in robotics, which necessitates superior generalization abilities. Opting for procedurally generated environments addresses this need by providing a far greater variety of data compared to photorealistic simulations, thereby increasing reinforcement learning models' robustness to unseen scenes, as demonstrated across various 2D (Cobbe et al., 2018; Johansen et al., 2019; Cobbe et al., 2019) and 3D (Jaderberg et al., 2018; Juliani et al., 2019) environments.

NavIndoor provides automatic generation of both semantic segmentation maps and depth maps as domain-agnostic features within procedurally generated environments, without the need for additional annotations or being constrained by a finite number of environments, contrary to existing platforms (Yadav et al., 2022) (Szot et al., 2021). Also, its design is specifically oriented for blind digital twin navigation including automatic generation of labeled collision instances. NavIndoor also includes parametrization for both agents (movement physics, action space, sensors) and environments (size, obstacle filling) and allows very high-speed rendering by leveraging various optimization features.

## 3 MATERIALS AND METHOD

We propose a scalable approach that leverages virtual environments, reinforcement learning, and transferable features to extract navigation-related features for SSDs, as illustrated in Figure 2. The navigation task is initially learned within a virtual environment by a digital twin equipped with sensor(s) and implemented through a reward function.

We used semantic segmentation masks as input for learning blind navigation within virtual environments, because they have lower dimensionality compared with depth maps, but still capture enough information for allowing navigation as shown in robotics (Hong et al., 2018; Chaplot et al., 2020).

Collection of synthetic navigation data was performed within virtual environments in the form of $(s_t, a_t, r_t)$, where $s_t$, $a_t$, and $r_t$ represent a semantic view, memory of actions taken by a blind digital twin, and reward values at time-step $t$, respectively. The navigation task is learned through policy learning from the semantic segmentation maps. Leveraging reinforcement learning, we estimate a parametric model $Q_\theta(s_t, a_t)$ using Q-learning. The Q-value function represents the expected sum of future rewards an agent would achieve in state $s_t$, choosing action $a_t$, and navigating optimally thereafter. Following the training, we evalyated the performance of $Q_\theta$ with real-world states.

To show the effectiveness of our approach, we propose to learn obstacle avoidance from a single head-mounted visual sensor, aligning with current start-of-the-art (see Figure 1) but without extensive real-world data collection and annotation. Semantic segmentation maps were processed to regroup obstacles in a single class and processed as $1 \times 128 \times 128$ tensors to reduce input dimensionality. We also leveraged procedural generation, allowing for randomization of virtual environments, semantic-segmentation specific data augmentation and used a compact convolutional neural network for Q-value estimation (1.6M parameters). These features aimed at improve model robustness and allow extraction of relevant feedback in real-time in a sensor-lightweight and energy-efficient pipeline.

For synthetic data generation, we developed a virtual environment, NavIndoor, encompassing the necessary features, and in which training was performed (Section 4). The following section presents the experimental setup employed for learning obstacle avoidance, including design of reward function, virtual environment, data randomization mechanisms, model architecture, hyperparameters optimisation as well as the model's results within virtual setting.

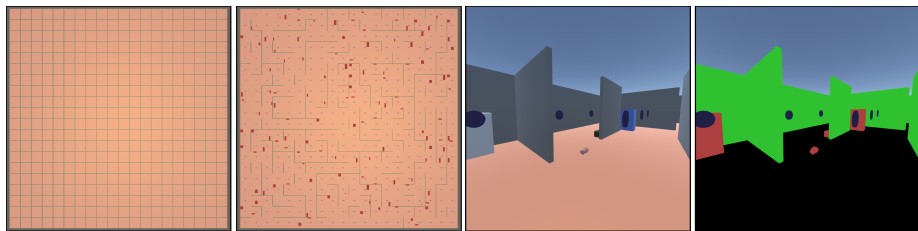

Figure 3: From left to right. The maze generation procedure initiates with a grid of closed cells and a randomly positioned agent. Subsequently, depth-first search algorithm is applied to its graph structure and cells are randomly filled with obstacles and collectibles. The agent wears a forehead RGB camera and a semantic segmentation camera.

# 4    TRAINING IN VIRTUAL ENVIRONMENT

## 3D ENVIRONMENT AND BLIND DIGITAL TWIN

NavIndoor is built upon Unity and MLAgents library (Juliani et al., 2018). It was developed as a platform for generation of synthetic data from navigation of a blind digital twin. NavIndoor is designed to create sequential, partially observable, static, procedurally generated environments filled with walls, obstacles, and collectibles. The environment generation includes maze generation and the sampling of a random starting point where the agent begins its exploration, in order to prevent memorization biases (Zhang et al., 2018). The generation procedure for mazes is based on the Depth First Search algorithm (Tarjan, 1972) and depicted in Figure 3. Obstacles include both cuboids of various shapes and open-source low-polygons assets. The environment was designed in Unity to allow for easy parametrization integration through Python for generation of mazes, agents and sensors with different properties.

A blind digital twin was designed to navigate these mazes. It has a discrete action space $\mathcal{A} = \{$forward, backward, rotate left, rotate right$\}$. The agent is equipped with a frontal monocular camera hat has a field of view (FOV) of $115° \times 100°$. The camera sensor is configured to return $128 \times 128$ semantic segmentation maps along with RGB views of the current scene.

## REWARD DESIGN AND OBSERVATIONS

We propose using collectibles located at the center of the maze's cells to design the reward function. The agent receives a positive reward when collecting a coin, which encourages exploration of the maze. It receives a negative reward when colliding with a wall or an obstacle. At each timestep, the environment returns a semantic map of the current view $seg_t$ as well as an RGB view $f_t$. The semantic map has five labels: *floor, obstacles, walls, coins, other*. At timestep $t$, the state $s_t = (seg_{t-2}, seg_{t-1}, seg_t, (a_i)_{i \in [t-m, t-1]})$ is extracted, where $seg_i$ is the semantic segmentation map at timestep $i$, $a_i$ is a one hot encoding of action taken at timestep $i$, and $m$ is an action memory length parameter. Encoding semantic maps as 2D arrays by associating a single value to walls (-0.5), floor (0.5), and obstacles (-1) provided better stability during training compared to multi-channel semantic encoding. Additionally, our experiments showed that providing time context through previously performed actions allowed the agent to escape situations where it would get stuck during learning, leading to better obstacle avoidance strategies.

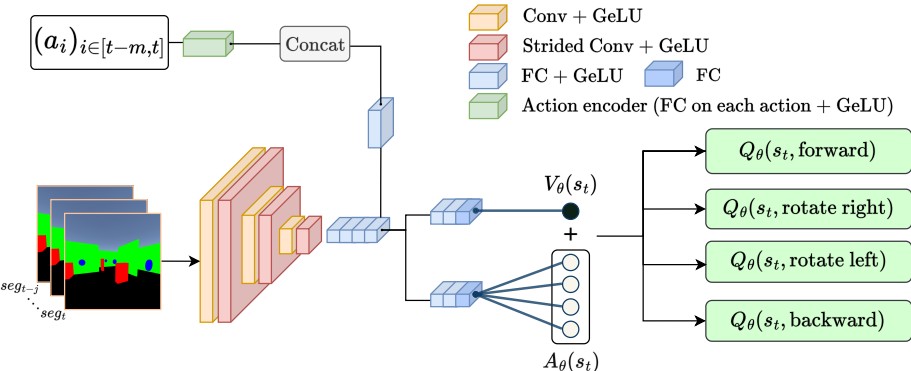

Figure 4: The Q-network architecture consists of six convolutional layers, followed by fully connected layers to estimate $Q_\theta(s)$. Given the state $s_t$, the deep Q-network estimates both the value and advantage functions, which can be interpreted as indicators of safety and guidance on potential actions, respectively..

## TRAINING

Obstacle avoidance was learned using Double Dueling Deep Q Network (D3QN) (Wang et al., 2015), with each episode taking place in a new procedurally generated maze. The model architecture processing $s_t$ is presented in Figure 4. It is composed of image and actions encoders followed with fully connected layers. The D3QN architecture offers an estimation $V_\theta$ of expected sum of discounted rewards, which relates directly to the safety level for obstacle avoidance. An extensive grid search was conducted over training hyperparameters and then agent and environment parameters (speed, obstacle proportion, progressive increase in difficulty) to determine the best training setting. Details on training and grid search are given in Appendix B. The best trained model (VC) setup was then used to trained another model (VCD) using data augmentation. Data augmentation included erosion and/or dilatation morphological operators around obstacle shapes using a fixed 3x3 square kernel with probability $p = 0.2$ for each image and operator. Morphological augmentation was followed by random uniform changes in each pixel label with probability $p = 0.05$. These changes aim to simulate potential errors during semantic segmentation of real-world images and to make the model robust to various obstacle shapes. Training setups included a model was trained from standard RGB views of the camera and a model trained with unvisible collectibles (NVC).

Two human individuals were also trained to collect the maximum amount of coins while avoiding collisions in to assess human performance and compare it to our models. Human performance was evaluated under settings with both visible and invisible collectibles, as well as RGB inputs, with 10 episodes for each individual. Each episode lasted for 24 seconds (corresponding to 400 decision timesteps) and humans navigated in the NavIndoor using the keyboard's directional arrows.

RESULTS IN NAVINDOOR

Table 1: Mean reward for final models *vs.* humans

|  | Best Model | Humans | Ratio |
|---|---|---|---|
| Visible coins | **74.34** (VC) | **102.36** | **0.73** |
|  | 73.18 (VCD) | / | 0.71 |
| Invisible coins | 47.44 (NVC) | 83.82 | 0.56 |
| RGB views | 63.98 | 101.34 | 0.63 |

Results after training are presented in Table 1. The best-trained model (VC) reached 73% of human-level performance. It had a mean reward of 74.34 and was obtained using visible collectibles and progressive increase in difficulty during training. The model trained with RGB views achieved a mean reward of 57.73, which was the lowest among models trained using semantic segmentation maps, showcasing the relevance of semantic segmentation for obstacle avoidance during visual-based navigation.

## 5 RESULTS

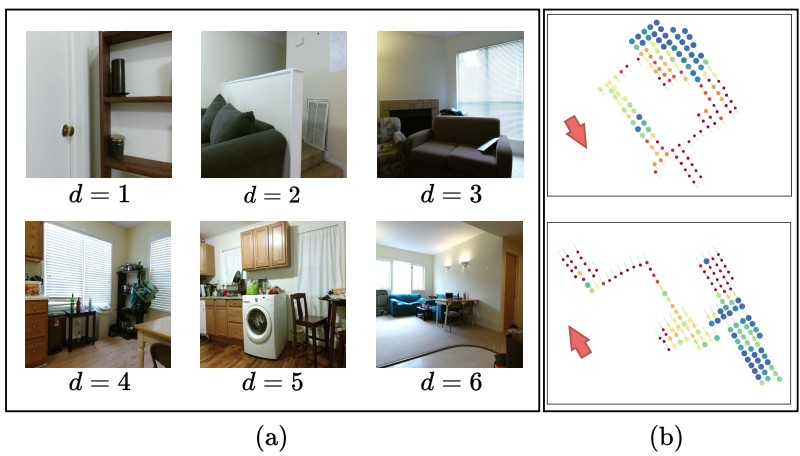

(a)                                        (b)

Figure 5: (a) AVD samples showing various distances to the navigation boundary. (b) A top-down view of two AVD rooms. Points mark the locations where photographs were captured. $Q_\theta(s, \text{forward})$ was computed for each state $s$, indicating the same direction (orange arrows). Blue (big) points indicates high values of $Q_\theta(s, \text{forward})$ and red (small) points low values.

We identified the Active Vision Dataset (AVD) as a good dataset for evaluating our model, because it offers indoor views coupled with spatial metadata, including coordinates and viewpoint angles for each navigable location. We computed for each image its distance to navigation boundary $d$, representing the number of possible forward steps from a given view. This value can be seen as a path clearance level. Samples from the AVD are illustrated in Figure 5 (a).

We conducted image segmentation as a pre-processing step using SegFormer-b2 on each AVD image. Next, we extracted features from VC and VCD models trained in NavIndoor using the initial state configuration (no previous action in the action memory buffer, and stacking the 3 same images). The process ran at 179 FPS in our setting (RTX 4090) without further optimization.

The model's output for VC correlated well with the path clearance level. In particular, the safety level $V_\theta(s)$ and the rotation advantage relative to going forward, $R_\theta(s)$, defined as the difference between

$Q_\theta(s, \text{forward})$ and the maximum between $Q_\theta(s, \text{rotate left})$ and $Q_\theta(s, \text{rotate right})$, correlated well with the path clearance level $d$ as depicted in Figure 6. We observed a mean increase of $V_\theta$ with respect to $d$, indicating the model's ability to assign better values to states with clearer pathways. Similarly, we noticed a mean decrease of $R_\theta(s)$ with respect to $d$, suggesting its potential as a navigation insight to indicate when rotfation is a favorable option. Additionally, we generated schemes of two AVD scenes and colored photograph locations for a specific direction based on $Q_\theta(s, \text{forward})$ in Figure 5 (b), showcasing high correlation with the forward pathway clearance.

The same processing method was applied on a video captured by a sighted individual wearing a camera on their forehead, and results showcased a high correlation between the forward distance to walls or obstacles with $Q_\theta(s)$ and $V_\theta(s)$. As depicted in Figure 8, $V_\theta(s)$ decreases when the individual approached obstacles, showcasing features relevance for feedback integration in real sensory substitution devices.

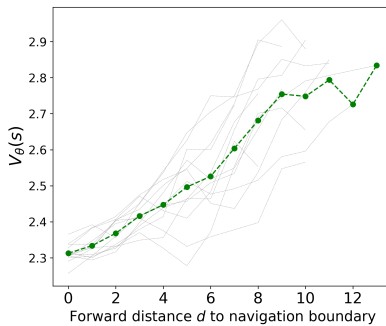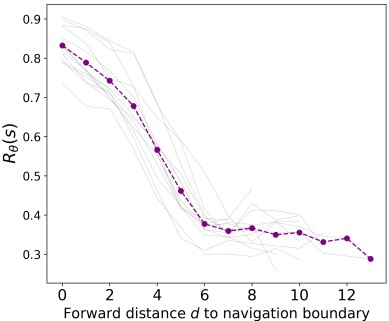

Figure 6: (Left) Mean values of $V_\theta$ on images within each scene (gray) and the overall average (green) are plotted with respect to $d$. (Right) Mean values of $R_\theta$ on images within each scene (gray) and the overall average (purple) are shown with respect to $d$ (left). Both $V_\theta$ and $R_\theta$ are single-dimensional features that significantly correlate with the distance to the navigation boundary in the real-world.

LINEAR PROBING

Quantitative evaluation was conducted through linear probing, involving binary classification of images based on their forward navigation boundary. Labels $y_i^d$ were determined by applying a threshold to the maximum reachable forward distance before encountering a wall or an obstacle. Specifically, $y_i^d$ was set to 1 if $d_i > d$, indicating a boundary within the next $d$ forward steps. This approach simulates the need to alert users when they approach obstacles with a binary feedback with alert distances $d$ varying from 0 to 6. Linear classifiers were trained on 9 indoor scenes and tested on 5 unseen scenes from different buildings (details in Appendix C).

Performance comparisons were made with other state-of-the-art models, including self-supervised backbones (DINOv2 distilled (Oquab et al., 2024), ConvNext V2 (Woo et al., 2023)) and supervised models (SegFormer-b2 (Xie et al., 2021), EfficientNet-b7 (Tan & Le, 2019)). For transformers models, the latent space of every image patch was used because it gave better performances compared with using only the *cls* token.

The results for AUCs of each classifier are depicted in Figure 7. Table 2 presents the mean evaluation metrics accross all trained classifiers. VCD classifiers consistently outperformed the other models for $d > 2$. Indeed, the use of morphological operators coupled with random changes in the unified semantic segmentation images thus provided with significantly better generalization with real-world semantic segmentation maps, which are naturally prone to errors. Although evaluation of VCD relied on pre-processing using SegFormer-

Table 2: Linear probing binary classifiers for forward navigation boundary detection. Mean metrics on test set.

| Model | Features | F1 | AUC |
|---|---|---|---|
| ConvNext V2 (Woo et al., 2023) | 15680 | 0.63 | 0.82 |
| SegFormer-b2 (Xie et al., 2021) | 131072 | 0.72 | 0.87 |
| EfficientNet-b7 (Tan & Le, 2019) | 231040 | 0.71 | 0.80 |
| DINOv2-B (Oquab et al., 2024) | 197376 | 0.75 | 0.86 |
| (Ours) VCD | **768** | **0.77** | **0.88** |
| (Ours) VC | **768** | 0.74 | 0.86 |
| (Ours) NVC | **768** | 0.69 | 0.85 |

b2, the results show significantly higher AUCs using VCD for $d > 3$ and higher F1 score, demonstrating our model performance in understanding structure of real-world indoor places.

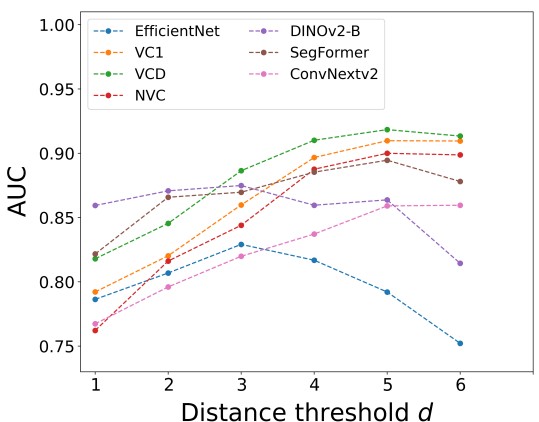

Figure 7: AUCs for different distance thresholds (binary classification) on test set for state-of-the-art models and VC, NVC, VCD.

Figure 8: Value function estimate $V_\theta$ across the real-world video sample. Image samples from the video are displayed with their associated $V_\theta$ (top arrow) and $A_\theta$ (bottom arrows) outputs.

## 6  CONCLUSION

In this study, we introduced a new framework aimed at improving mobility for visually impaired people through the use of synthetic data. We proposed a virtual environment specifically designed for generating human-like navigation data, which can be used for training and evaluating deep learning models for SSDs. Compared to previous approaches, our method offers scalability and real-time extraction of low-dimensional features for safe navigation from a single visual sensor. Indeed, the proposed method ensures a high compatibility with the lightweightness, cognitive and hardware constraints associated with SSDs. These advances represent a significant step in the development of robust AI-based assistive technologies and pave the way for further research aimed at improving mobility for visually impaired individuals.

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

## A  LITERATURE REVIEW

To assess the relevance of AI methods for sensory substitution devices, we conducted a literature review on sensory substitution devices and electronic travel aids published since 1970. The review was performed in PubMed using binary strings that combined one sensory substitution-related term (e.g., sensory substitution, electronic travel aid) with a signal-processing-related term (e.g., deep learning, computer vision, machine learning, image processing, signal processing, artificial intelligence, neural networks). The results were supplemented with research from Elicit, various search engines, and arXiv. We included papers proposing new systems for sensory substitution, resulting in a total of 61 studies. Each paper was analyzed and labeled according to the information detailed in Table 3, including the type of image or signal processing used, as well as additional information such as the device's purpose, type of sensors, and sensor location. The full review results are presented in Tables 4,5.

| Reviewed Property | Description and acronyms |
|---|---|
| Year | Publish year. |
| Sensor (S) | Sensors used for environment information acquisition. *RGB Camera (CAM), RGB-Depth Camera(s) (D-CAM), Ultrasonic (US), LIDAR, Infrared (IR), Externally added Geographic Information (GI), GPS, Laser Beam (LB).* |
| Purpose | Purpose of the SSD. *Generic Use for any application (GU), Obstacle Avoidance (OA), Navigation (NAV), Object Recognition (OR), Localization (LOC).* |
| Processing Tools (TOOLS) | Methodological Tools used for information extraction. *Neural Networks (NN), Computer Vision (CV), Machine Learning traditional approach (ML), Signal Processing (SP).* |
| Sensor Location (SLOC) | Location of the acquisition sensors. |

Table 3: List of reviewed aspects for prosthetic vision studies and their acronyms

| Ref | Year | S | Purpose | Tools | SLOC |
|---|---|---|---|---|---|
| Collins (1970) | 1970 | CAM | GU | CV | back |
| Meijer (1992b) | 1992 | CAM | GU | CV | NA |
| Bach-y Rita et al. (1998) | 1998 | CAM | GU | CV | NA |
| Shoval et al. (1998) | 1998 | US | OA | SP | belt |
| Rodríguez-Ramos (2009) | 1999 | D-CAM | GU | CV | glasses |
| Arno et al. (1999) | 1999 | CAM | GU | CV | Head |
| Ulrich & Borenstein (2001) | 2001 | US | OA | SP | A-Cane |
| Kuc (2002) | 2002 | US | OA,OR | SP | NA |
| Segond et al. (2005) | 2005 | CAM | OA | CV | abdomen |
| DANILOV & TYLER (2005) | 2006 | CAM | GU | NA | head |
| Kajimoto et al. (2006) | 2006 | CAM | GU | CV | head |
| Johnson & Higgins (2006) | 2006 | D-CAM | OA | CV | belt |
| Dakopoulos et al. (2007) | 2007 | D-CAM | OA | CV | glasses |
| Tekin & Coughlan (2009) | 2009 | CAM | OR | ML | NA |
| Coughlan & Manduchi (2009) | 2009 | CAM,GI | WF | CV | hand |
| Winlock et al. (2010) | 2010 | CAM | OR | ML | NA |
| Khan et al. (2012) | 2012 | D-CAM | OR,OA | CV | belt |
| Rodríguez et al. (2012) | 2012 | D-CAM | OA | CV | chest |
| Vera et al. (2014) | 2013 | CAM,LB | OA | SP | hand |
| Tapu et al. (2013) | 2014 | CAM | OA | ML | chest |
| Elloumi et al. (2013) | 2014 | CAM,GI | WF | ML | chest |
| Maidenbaum et al. (2014) | 2014 | IR | OA | SP | hand |
| Kajimoto et al. (2014) | 2014 | CAM | GU | CV | hand |
| Garcia & Nahapetian (2015) | 2015 | CAM | OA | CV | glasses |
| Zheng & Weng (2016) | 2016 | CAM,GPS | OA | NN | hand |
| Bulat & Glowacz (2016) | 2016 | D-CAM | OA | CV | NA |
| Ivanchenko et al. (2008) | 2016 | CAM | OR | ML | hand |
| Schwarze et al. (2015) | 2016 | D-CAM,IS | OA | CV | helmet |
| Kerdegari et al. (2016) | 2016 | US | OA | NN | helmet |
| Ko & Kim (2017) | 2017 | CAM,IS,GI | WF | ML | hand |
| Rituerto et al. (2016) | 2017 | CAM,IS,GI | LOC | CV | chest |
| Tapu et al. (2017) | 2017 | CAM | OA | NN | belt |
| Bai et al. (2017) | 2017 | D-CAM, CAM,GI | NAV | NN | helmet |
| Sharma et al. (2016) | 2017 | CAM | OR,OA | NN | chest |
| Agarwal et al. (2017) | 2017 | US | OA | SP | glasses |
| Dasila et al. (2017) | 2018 | D-CAM | OR,OA | CV | NA |

Table 4: Review Results part.1

| Ref | Year | S | Purpose | Tools | SLOC |
|---|---|---|---|---|---|
| Caraiman et al. (2017) | 2018 | IR,D-CAM | OA,OR, TR | ML | abdomen |
| Mancini et al. (2018) | 2018 | CAM | WF | CV | Gloves |
| Bhat et al. (2017) | 2018 | CAM | OR,TR | ML | NA |
| Lobo et al. (2019) | 2019 | IR | OA | SP | hand |
| Hu et al. (2019) | 2019 | CAM | GU | NN | NA |
| Rahman et al. (2020) | 2020 | US,IR, IS,GI | OA | SP | shin |
| Port et al. (2021) | 2020 | CAM | OR | NN | NA |
| Afif et al. (2020) | 2020 | CAM | OR | NN | None |
| Zhao et al. (2020) | 2020 | CAM,GPS | WF,OA | NA | glasses |
| Kavya & G C (2020) | 2020 | CAM,US, GPS | OA,OR, TR | NN | A-Cane |
| Kim et al. (2021) | 2021 | CAM | GU | NN | glasses |
| Dernayka et al. (2021) | 2021 | LIDAR,IR | OA | SP | A-Cane |
| Mukhiddinov & Cho (2021) | 2021 | CAM | OR,TR | NN | NA |
| Wright & Ward (2013) | 2021 | CAM | GU | ML | NA |
| Busaeed et al. (2022) | 2022 | LIDAR,LB, US,GPS | OA | NN | glasses |
| Ganesan et al. (2022) | 2022 | CAM | OR | NN | NA |
| Roy & Shah (2022) | 2022 | GPS,US | OA,LOC | SP | A-Cane |
| Bhatlawande et al. (2022) | 2022 | CAM,US | OR,OA | NN | Hand |
| Scalvini et al. (2023) | 2023 | D-CAM, GPS,IS | NAV | NN | helmet |
| Asiedu Asante & Imamura (2023) | 2023 | D-CAM | OA | NN | abdomen |
| Kim et al. (2023) | 2023 | CAM | GU | NN | NA |
| Sulaman et al. (2023) | 2023 | CAM,US, GPS,IF | OA,OR, LOC | NN | A-Cane |
| Chaudhary & Dr. PrabhatVerma (2023) | 2023 | CAM | OR | NN | None |
| Kavitha et al. (2023) | 2023 | CAM | OR | NN | None |
| Kumar et al. (2023) | 2023 | CAM,US, GPS,IS | OA,OR, LOC | CV | A-Cane |

Table 5: Review Results part.2

## B GRID SEARCH IN NAVINDOOR

Training was conducted alongside the agent's exploration of the environment, utilizing an $\epsilon$-greedy policy.

Each episode took place in a new maze and lasted for 400 decision timesteps, for both training and evaluation. At a fixed frequency, data was sampled from a replay buffer and used for $Q_\theta$ parameters update. Semantic maps used for training were encoded as 2D arrays with values: 1 for coins, 0.5 for floor, $-0.5$ for walls, $-1$ for obstacles, and 0 elsewhere. Training lasted for 250 episodes. The model was implemented with PyTorch and trained using NVIDIA RTX 4090.

Obstacle avoidance was learned using Double Dueling Deep Q Network (D3QN) Wang et al. (2015) which the associated loss function $L$ is given by:

$$L(\theta) = \mathbb{E}\left[(y - Q_\theta(s, a))^2\right], \tag{1}$$

where the target value $y$ for the Double Q-Learning approach is defined as:

$$y = r + \gamma Q_{\theta^-}\left(s', \underset{a'}{\arg\max} \, Q_\theta(s', a')\right), \tag{2}$$

and in the Dueling Network architecture, the Q function is decomposed into:

$$Q_\theta(s, a) = V_\theta(s) + \left(A_\theta(s, a) - \frac{1}{|\mathcal{A}|} \sum_{a'} A_\theta(s, a')\right), \tag{3}$$

where $V_\theta(s)$ represents the value function, $A(s, a; \theta)$ the advantage function, $|\mathcal{A}|$ the number of possible actions, $\theta$ and $\theta^-$ the parameters for current and target networks, respectively.

Considering the complexity of reinforcement learning in 3D environments, we performed a grid search to understand the model's sensitivity to training parameters. The parameters considered for the grid search, along with their associated **best** values, are as follows: model size (1.1, **1.6M**, 2.4 parameters), learning rate (0.0001,0.0005,**0.001**), batch size (64,**128**,256), discount factor $\gamma$ (0.95,**0.97**,0.99), action memory length $m$ (10,**20**), $\epsilon_{min}$ (0.15,**0.3**), training proportion of linear decrease ratio for $\epsilon$ (0.25,**0.33**,0.5), training iteration frequency (**5**,10), coin reward value (1,**5**,10), update type of $Q_{\theta^-}$ (***hard* update every 50 training steps**, *hard* update every 10 training steps, *soft* update). This grid search, as well as all evaluation results use a default setting specified below. After the grid search, we conducted training sessions with the best-found hyper-parameters. A total of 16 training sessions were performed, considering 4 binary settings, and the best models for visible/invisible collectibles were designated as VC/NVC, respectively. Then, using the VC settings, 2 additional training sessions were conducted to train the RGB model with data augmentation for VCD.

The first binary setting examined fixed moving speed (default, fixed to 1) *vs* random moving speed (sampled between 0.75 and 1.25) to assess the model's ability to generalize to agents with different speeds. The second binary setting investigated the need for visual cues by training with visible (default) *vs* invisible coins. Visual cues assist the model in navigation but also introduce a bias for generalization to real-world semantic segmentation maps, which may not include such visual cues. The third binary setting implemented decreasing rewards only at collision time (default) *vs*decreasing rewards when staying near an obstacle. This study aims to determine if this approach could prevent situations where the agent becomes stuck by colliding with an obstacle.

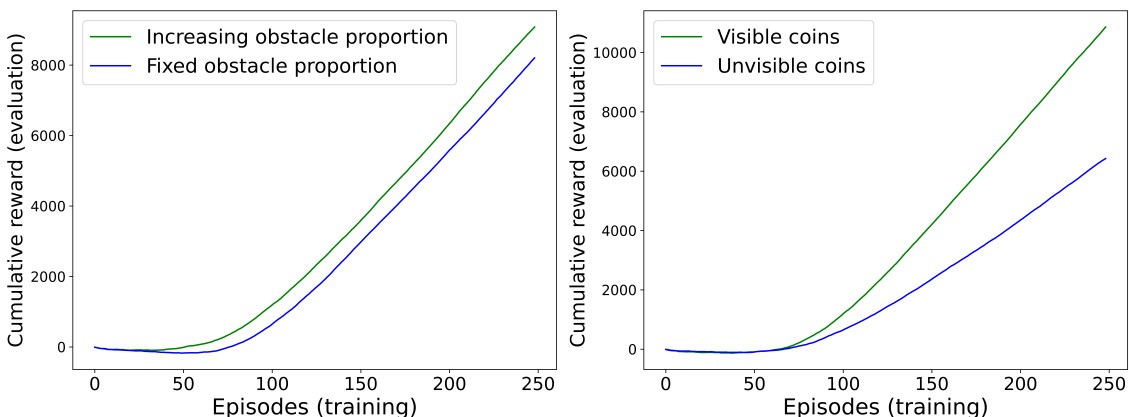

Figure 9: Mean cumulative rewards are shown for fixed *vs* variable difficulty during training (left) and for visible *vs* invisible coin settings (right). The mean cumulative reward was computed over the 8 models in each group.

The last binary setting aimed to study potential improvements in performance by progressively increasing task difficulty through obstacle proportion, in accordance with previous studies Kulhánek et al. (2019); Justesen et al. (2018). We compared a fixed proportion of obstacles throughout training (default) with a linear increase in obstacle proportion during training. Following training, each of the 16+2 models was evaluated on 500 unseen mazes under the default setting.

Two subgroups showed significant differences compared to their binary counterparts. First, the coin visibility group demonstrated better performance (mean reward of 72.76), while the model still exhibited learning capabilities even without visual cues (mean reward of 45.22). Although the gradual increase in difficulty did not significantly enhance the mean reward, it accelerated the convergence of the models. These results can be observed in the cumulative rewards of these subgroups throughout training, as shown in Figure 9. Cumulative rewards and mean rewards after training showed no significant differences for the other two binary settings.

## C  TRAIN/TEST SPLIT IN ACTIVE VISION DATASET

Linear probing was performed using the currently available data from the Active Vision Dataset Houses. Scenes from Houses 1, 2, 3, 4, and 7 were used for training, while Houses 10, 11, 13, 15, and 16 were used for testing.

