# OpenReview forum: "Reinforcement Learning on Synthetic Navigation Data allows Safe Navigation in Blind Digital Twins"
_ICLR.cc/2025/Conference — ICLR 2025 Conference Withdrawn Submission_

### Official Review · Reviewer_aN9A · 2024-10-31

**Soundness:** 1
**Presentation:** 3
**Contribution:** 1
**Rating:** 1
**Confidence:** 4

**Summary:**

The goal of this paper is to propose and evaluate a methodology for learning a vision model that can act as a sensory substitution device for the blind/visually impaired. The paper proposes a method for generating simulation data, which is then used to train a model. This model determines where it is safe to travel, specifically forward, left, right, or backward. The model takes as input a semantic segmentation of the scene and the action history, and outputs the Q-value of each of the 4 actions. The results show that the best model can achieve nearly 75% human performance in simulation, and on the real dataset  the output of the value function is strongly correlated with distance to navigation boundary. This suggests that it can act as a good aide for a visually impaired person moving through an indoor space.

**Strengths:**

*This paper has the most thorough literature review I have ever seen in a conference paper (109, I counted! As many pages of citations as paper). I in particular really appreciate Figure 2 and the comprehensive classification of prior work on sensory substitution devices. This is great at giving a background on SSDs to an audience which may be more familiar with ML.

*The motivation behind this paper is excellent. Helping the visually impaired see and navigate through human environments is incredibly important, and this paper does an excellent job of establishing why we should care about this work.

*There are many papers in AI/ML/robotics dedicated to sim2real transfer, but in this paper it works fairly well. The simulation is fairly simplistic (at least visually) and the real evaluation environment uses images taken in the real world, so the transfer is no easy feet. The use of semantic segmentation masks as input to the model likely makes a big difference, however even so it is still non-trivial. I am surprised at the transfer success.

*In the Linear Probing section, there is a comparison to several state of the art baselines. Comparing to baselines is incredibly important, so the comparison here is a good thing.

**Weaknesses:**

*The biggest weakness with this paper is that the results are just not convincing. The numbers presented are 1) comparison to human-performance on the sim task (human having full visual sight), 2) distance to navigation boundary as compared to output of the value function, and 3) AUC comparison to the baselines (note that AUC is never defined in the paper, and the abbreviation is never clarified (I assume it stands for area under the curve?)). If the goal is to assist visually impaired people navigate, how do these numbers show that?
They are very indirect. Why not try actual navigation tasks with the proposed model and baselines? To convince me this model is actually useful, I need to see more directly applicable results.

*The paper claims the value function is an indicator of safety and guidance (Figure 4 caption). This is not well-justified. Why should the arbitrary “best possible reward the agent can get from a given state” be the same as whether or not that state is safe?

*There are several missing citations. SegFormer is never cited. Furthermore, the Active Vision Dataset is not cited. Where did it come from? Who collected it? What kind and how much data is there? How are you estimating distance to a navigation boundary?

*The creation of the simulated data is not explained well enough. It seems only the Figure 3 caption gives any data on this, and that is not much. How is DFS used to place the obstacles and collectibles?

*Also, why are there collectibles? That seems like a random addition to the simulated data. If your goal is to get the agent to explore, there are many ways in the RL literature to motivate exploration (e.g., maximum entropy).

**Questions:**

This is more advice than question, but the biggest thing this paper can do to increase its quality is to generate more convincing results. The easiest thing to try would be to put a camera on blindfolded real human participants and have them use the proposed models and baselines and see which allows them to avoid obstacles the best. Alternatively (if the IRB approvals for that are too hard to get), train an RL model that takes as input exactly the same sensory substitution that a visually impaired person would get from the model, and report how well it does at navigation tasks, both with the proposed model and the baselines.

---

### Official Review · Reviewer_erv3 · 2024-11-02

**Soundness:** 1
**Presentation:** 1
**Contribution:** 2
**Rating:** 3
**Confidence:** 5

**Summary:**

- The authors developed a virtual environment designed to extract various human-like navigation data from procedurally generated labyrinths.

- Using reinforcement learning and semantic segmentation, authors trained a convolutional neural network to perform obstacle avoidance from input RGB data. They demonstrated that their model outperformed state-of-the-art backbones including DINOv2-B in safe pathway identification in the real world.

**Strengths:**

1. The authors talk in their article about the need to solve an extremely important and sensitive problem of creating intelligent navigation systems for visually impaired individuals.

2. The authors developed and tested NavIndoor, an open-source software for the computationally efficient generation of procedurally generated, obstacle-filled environments, enabling seamless integration with AI systems.

**Weaknesses:**

1. The overview of methods in Figure 1 requires some improvement: in the subfigure named "Signal processing method", the authors mention Machine Learning and Neural Networks separately, but Neural network training is also machine learning. The category "Of which trained using blind specific data" also looks strange.

2. The Related work section does not contain a single paper from 2024, which is strange. It is necessary to explicitly indicate that there are no such works, or add them to the overview.

3. The Q-network architecture proposed by the authors is very simple and it is unclear how it differs from existing models used in modern works on reinforcement learning. The authors should add an explicit mention of the differences in the caption to Figure 4. Please, compare specific aspects of author's architecture to existing models, or to highlight any novel elements that may not be immediately apparent from the figure.

4. On page 8, Active Vision Dataset (AVD) is mentioned, but no reference to the source is provided. The authors need to explain what this dataset is.

5. Figure 8 does not have labels for the values ​​on the vertical axis. They should be added.

6. In the abstract and introduction, the authors say that their system and method are specifically developed for visually impaired individuals. However, the developed dataset, methodology, and experiments look as if they are solving a general navigation problem typical for intelligent agents (robots, etc.) using data from onboard sensors with a discrete action space. The authors need to explicitly clarify this in the article. Otherwise, the title of the article may mislead the reader.

7. The usefulness of the developed solution is questionable due to the poor photorealism of the simulator used and the overly simplified formulation of the navigation problem. Photorealism is generally important for the quality of image-based navigation methods to effectively transfer to real-world environments.

8. The English language of the article requires careful checking, for example, "in" does not look entirely correct in the phrase "dedicated navigation data in visually impaired individuals" in the abstract. The text contains unnecessary punctuation marks, for example, several dots in a row. Typo "developed" in the abstract.

**Questions:**

1. Why didn't the authors use photorealistic simulators Habitat Sim and AI2Thor, which can solve indoor navigation problems, to train and validate their approach?

2. Could the authors explain their rationale for developing NavIndoor rather than using existing environments like Habitat or AI2Thor? Are there specific advantages of NavIndoor for this task that are not provided by these other environments?

3. Have the authors considered comparing their approach to more recent reinforcement learning methods, such as those used in Staroverov, A., et al. "Skill fusion in hybrid robotic framework for visual object goal navigation." Robotics 12.4 (2023): 104? What specific benchmarks or challenges do they think would be most relevant for evaluating their system's performance?

4. Why didn't the authors include any form of anonymized open source code in the article or supplementary materials?

---

### Official Review · Reviewer_C83X · 2024-11-03

**Soundness:** 2
**Presentation:** 2
**Contribution:** 2
**Rating:** 3
**Confidence:** 4

**Summary:**

This paper describes a RL method using synthetic data to train a model that can assist blind people in navigating through real-world environments.

**Strengths:**

The basic idea/concept of using synthetic datasets to train a learning model for navigation is fine and has in fact been done in some of the latest work on model constructions in CV/ML and SLAM in learning-based robotics, as well as many other CV/ML applications for learning-based robotics tasks (e.g. navigation, collision avoidance, driving/steering, etc).

**Weaknesses:**

The motivation about suggesting/implying to provide some forms of assistive tools for blind people seems completely irrelevant to the work presented in the paper.   This paper presents a method to train a model for indoor navigation using semantic segmentation. There is no clear explanation on how the method is actually to be used by blind people as assistive devices for blind people to use for navigation (sensory substitution devices, as the paper claims).  I suggest the authors to clarify the connection between the proposed work and the stated application.   Please clearly explain how the semantic segmentation and navigation model could be integrated into an actual sensory substitution device.  Please also discuss specific requirements of assistive technologies for the visually impaired that could inform them how to use the proposed research.

I did not find the results to be particularly illuminating or superior, considering that the performance is more or less in the same range as existing SLAM algorithms for relatively controlled and simple environments.

Examples shown also do not indicate any generalization capability to me.

The description also does not offer any rationale for high robustness either.

**Questions:**

- Why did the authors not just use one of the many many indoor environment datasets that are already available (e.g., RoboThor. Matterport3D, Apple's Hypersim, Meta's Ego-Exo4D). Instead, their simulation framework has very very low visual realism which will affect the performance, when moving to real-world video data. Why not just train the model using one of these datasets that has high-quality realism?   I'd suggest that the authors compare their approach using their custom environment to results using one or more of the existing datasets mentioned above. Additionally, I'd suggest the authors to discuss the specific advantages of their NavIndoor environment compared to existing datasets, particularly in relation to training navigation models for visually impaired users.   User studies should be conducted in these comparisons.

- Why did the authors not conduct any user study with blind people, so I'd consider the results pretty useless/irrelevant to the target user groups claimed in the title?   I'd suggest that some user studies/evaluations with visually impaired participants for navigating in real world with and w/o this approach to strengthen the paper's claims and relevance to the target application.

- There is not enough explanation about what a Sensory Substitution Device is, and this is the entire motivation of their paper.  I suspect that most readers in the ICLR community would not know without explanation what a Sensory Substitution Device (SSD) is. I'd suggest that the authors add a dedicated subsection in the introduction or background to define SSDs and explain their relevance to the proposed research, and discuss how neural networks can be applied and used in this context clearly -  possibly with some diagrams to illustrate the use case.

- While the basic idea of the described approach is not flawed (train a model to do environment navigation, use that model to help users navigate in the real world), the proposed approach don't seem to offer any novelty over a large body of learning-based model construction and SLAM literatures, as well as many similar works published in robot mapping literatures.  I'd suggest the authors compare with some of the latest works on SLAM or robot mapping techniques [1],  Additionally, authors can more clearly articulate the novel aspects of their method in the context of existing literature on learning-based navigation against other mapping and SLAM

- Can the authors perform an extensive comparison and user studies to justify against a large body of existing work on SLAM (see a recent survey in [1]) in the same context for guiding the visually impaired users?  Using a semantic segmentation model to navigate through a (synthetic or real) indoor environment, and then evaluate such a model on pre-processed semantically-segmented images from a real-world indoor environment dataset?  I'd suggest the authors to conduct a thorough user study on the target group (i.e. visually impaired), to support any meaningful claim on the key contribution of this paper.

- In terms of robustness and generalization, what's the insight on why this approach would do any better than existing methods?  I'd suggest the authors to provide more empirical evidence and/or theoretical justification for why their synthetic data generation and training approach leads to better robustness or generalization compared to existing methods, such as those mentioned in [1].

[1] Deep reinforcement learning based mobile robot navigation: A review.  https://ieeexplore.ieee.org/abstract/document/9409758

**Details Of Ethics Concerns:**

- It's not okay to refer to blind people as "blinds" (line ~27)

- I find it a little dubious to use 'blind people' as some target user groups without ever getting any input or user evaluation
  from the target group.

---

### Official Review · Reviewer_G6da · 2024-11-03

**Soundness:** 2
**Presentation:** 2
**Contribution:** 1
**Rating:** 1
**Confidence:** 3

**Summary:**

The authors try to overcome limited real-world datasets or environments that are expensive to train by using simulators, but at the same time using limited real-world datasets to incorporate selective learning or domain transfer in the optimization pipeline.

**Strengths:**

Strengths:

1. The figures are easy and intuitive to understand.
2. The experiments performed are represented on visuals well.

**Weaknesses:**

Weakness:
1. The problem motivation is clear to me and the problem formulation is also clear to me. But I don’t understand the connection between two. I think both of them are independent problems, in the sense that the authors could’ve just directly posed it as a sim2real navigation problem, whereas visual navigation in the world is: Can’t it be done so? Please correct me if I’m wrong.
2. My research is in visual navigation, and from what I see, to put it briefly in words, this problem would have made a lot of sense 4-5 years ago when obtaining a real-world policy was expensive to learn in real-world environments. But with current sota visual navigation algorithms and realistic simulators, I think both algorithms trained in a wide variety of simulator data and models that have already incorporated lots of indicative knowledge and priors from large datasets (LLMs, VLMs, RTX, etc.) would generalize well to real-world tasks. I’d suggest the authors incorporate these models as the baselines instead of models that output some form of representations and then training policy on top of that.
3. From what I see, the authors need to spend a bit more time in the manuscript presentation, not that there are too many typos, but I think the formatting and sizes of different figures with the text is not coherent and consistent
4. The paper also has a lot of technical flaws in the experiment section; for example, the only strength I see is in table 2, but I fail to even understand what the full form of VCD is, let alone understand the technical aspects. I suggest the authors lay out their contributions well and elaborate on each of the specific contributions.

**Questions:**

I think there is quite some work that needs not only the experiments and organization of the paper but also reiterating in the problem formulation.

---

### Official Review · Reviewer_R7jj · 2024-11-04

**Soundness:** 2
**Presentation:** 2
**Contribution:** 2
**Rating:** 5
**Confidence:** 3

**Summary:**

The paper presents a method and a simulator for training sensory substitution devices aimed at facilitating safe navigation for visually impaired individuals. The authors employ reinforcement learning to handle obstacle avoidance tasks. Training is initially conducted in a simulated environment, where a semantic segmentation camera captures segmentation maps as input. For real-world application, an external segmentation model processes RGB images to generate similar segmentation maps, which are then fed into the navigation model. The authors design a simple, compact model that relies exclusively on segmentation maps, optimizing for computational efficiency. They evaluate the model’s performance both in simulation and on real-world datasets, comparing it with pre-trained state-of-the-art models.

**Strengths:**

- small and compact model purely trained on segmentation map, good computation efficiency.

- use synthetic data and follow a sim2real approach

- potential impact on visually impaired individuals.

**Weaknesses:**

- in abstract, line 015, typo "developped" should be "developed"

-  It might not be necessary to build a stand alone simulation platform to achieve the task, the scene creation and data collection can definitely be accomplished using existing simulators. Many available simulators focus on photorealism but could still be effective for this task, as they have already demonstrated good results in sim and real. Therefore, I'm wondering if it worth the effort to develop a simulator just for this task.

- segmentation categories are quite limited, which may not sufficiently capture the rich semantics required in complex, real-world environments.

- It might not be a fair comparison to compare the model specifically for navigation task with general-purpose models trained for recognition.

**Questions:**

- I’m curious about the decision to use a CNN and D3QN rather than exploring more advanced architectures. If on-device computation efficiency is a concern, have you considered using pruning and quantization to optimize more complex models?

- Have you tested the model on a portable device? The reported testing on an RTX 4090 at 179 FPS seems beyond the computational needs of the task and may not reflect real-world performance on a SSD, where hardware specifications vary significantly. Since efficiency was a key factor in selecting a simpler model structure, testing on a lower-power or portable device could provide more insight into practical deployment. Alternatively, a wireless solution might offer a way to handle intensive computation remotely, particularly in indoor settings where connectivity is reliable.

---

### Official Review · Reviewer_HodL · 2024-11-08

**Soundness:** 3
**Presentation:** 3
**Contribution:** 3
**Rating:** 6
**Confidence:** 4

**Summary:**

Limited access to specialized navigation data for visually impaired individuals remains a major obstacle in advancing AI-driven assistive devices. To address this challenge, this work introduces a virtual environment specifically designed to generate human-like navigation data from procedurally generated labyrinths. Utilizing reinforcement learning and semantic segmentation, a convolutional neural network was trained to perform obstacle avoidance based on synthetic data. The resulting model surpassed state-of-the-art backbones, including DINOv2-B, in accurately identifying safe pathways in real-world settings. Overall, despite training exclusively on synthetic data, the model successfully extracted features conducive to safe navigation in real-world conditions, potentially paving the way for innovative solutions to assist the visually impaired.

**Strengths:**

(1) A significant gap in the literature is identified concerning the use of navigation data to enhance Sensory Substitution Systems.
(2) NavIndoor, an open-source software, is introduced for the computationally efficient generation of procedurally generated, obstacle-filled environments, enabling seamless integration with AI systems. NavIndoor supports the efficient creation of large-scale, human-like navigation datasets.
(3) The study demonstrates that synthetic data enables the extraction of low-dimensional features for navigation by individuals with visual impairments.
(4) It is demonstrated that applying basic morphological operators to synthetic semantic segmentation maps enhances performance in real-world conditions after training.

**Weaknesses:**

(1) The model's training primarily relies on synthetic navigation data from procedurally generated environments, which may not fully capture the complexity of real-world conditions. This could introduce limitations in the model’s generalization to unpredictable real-world scenarios.
(2) While the paper claims some level of real-world transferability, there is limited discussion on effective domain adaptation techniques or extensive testing in real environments. This lack of robust domain adaptation could mean the model's performance may vary significantly when exposed to real-world conditions without sufficient adaptation.
(3) The study lacks discussion on how well the model’s navigation cues (such as haptic or auditory feedback) are perceived and utilized by visually impaired users. Testing on real users could provide valuable insights into how user-friendly and effective the system is in practical assistive scenarios.

**Questions:**

A lack of specialized navigation data for visually impaired individuals hinders progress in AI-driven assistive devices. To address this, the work presents a virtual environment that generates human-like navigation data using procedurally generated labyrinths. A convolutional neural network, trained with reinforcement learning and semantic segmentation on synthetic data, enables effective obstacle avoidance. But I have some concerns illustrated in the weaknesses section. Looking forward to seeing the response from the author for these questions.

---

### Note · Authors · 2025-01-15

I have read and agree with the venue's withdrawal policy on behalf of myself and my co-authors.